# Bayesian optimization explains human active search

**Ali Borji**
Department of Computer Science
USC, Los Angeles, 90089
borji@usc.edu

**Laurent Itti**
Departments of Neuroscience and Computer Science
USC, Los Angeles, 90089
itti@usc.edu

## Abstract

Many real-world problems have complicated objective functions. To optimize such functions, humans utilize sophisticated sequential decision-making strategies. Many optimization algorithms have also been developed for this same purpose, but how do they compare to humans in terms of both performance and behavior? We try to unravel the general underlying algorithm people may be using while searching for the maximum of an invisible 1D function. Subjects click on a blank screen and are shown the ordinate of the function at each clicked abscissa location. Their task is to find the function's maximum in as few clicks as possible. Subjects win if they get close enough to the maximum location. Analysis over 23 non-maths undergraduates, optimizing 25 functions from different families, shows that humans outperform 24 well-known optimization algorithms. Bayesian Optimization based on Gaussian Processes, which exploits all the $x$ values tried and all the $f(x)$ values obtained so far to pick the next $x$, predicts human performance and searched locations better. In 6 follow-up controlled experiments over 76 subjects, covering interpolation, extrapolation, and optimization tasks, we further confirm that Gaussian Processes provide a general and unified theoretical account to explain passive and active function learning and search in humans.

## 1 Introduction

To find the best solution to a complex real-life search problem, e.g., discovering the best drug to treat a disease, one often has few chances for experimenting, as each trial is lengthy and costly. Thus, a decision maker, be it human or machine, should employ an intelligent strategy to minimize the number of trials. This problem has been addressed in several fields under different names, including active learning [1], Bayesian optimization [2, 3], optimal search [4, 5, 6], optimal experimental design [7, 8], hyper-parameter optimization, and others. Optimal decision making algorithms show significant promise in many applications, including human-machine interaction, intelligent tutoring systems, recommendation systems, sensor placement, robotics control, and many more.

Here, inspired by the optimization literature, we design and conduct a series of experiments to understand human search and active learning behavior. We compare and contrast humans with standard optimization algorithms, to learn how well humans perform 1D function optimization and to discover which algorithm best approaches or explains human search strategies. This contrast hints toward developing even more sophisticated algorithms and offers important theoretical and practical implications for our understanding of human learning and cognition.

We aim to decipher how humans choose the next $x$ to be queried when attempting to locate the maximum of an unknown 1D function. We focus on the following questions: Do humans perform local search (for instance by randomly choosing a location and following the gradient of the function, e.g., gradient descent), or do they try to capture the overall structure of the underlying function (e.g., polynomial, linear, exponential, smoothness), or some combination of both? Do the sets of sample $x$ locations queried by humans resemble those of some algorithms more than others? Do humans follow a Bayesian approach, and if so which selection criterion might they employ? How do humans balance between exploration and exploitation during optimization? Can Gaussian processes [9] offer a unifying theory of human function learning and active search?

## 2 Experiments and Results

We seek to study human search behavior directly on 1D function optimization, for the first time systematically and explicitly. We are motivated by two main reasons: 1) Optimization has been intensively studied and today a large variety of optimization algorithms and theoretical analyses exist, 2) 1D search allows us to focus on basic search mechanisms utilized by humans, eliminating real-world confounds such as context, salient distractors, semantic information, etc. A total of 99 undergraduate students with basic calculus knowledge from our university participated in 7 experiments. They were from the following majors: Neurosciences, Biology, Psychology, Kinesiology, Business, English, Economics, and Political Sciences (i.e., not Maths or Engineering). Subjects had normal or corrected-to-normal vision and were compensated by course credits. They were seated behind a $42"$ computer monitor at a distance of 130 cm (subtending a field of view of $43° \times 25°$). The experimental methods were approved by our university's Institutional Review Board (IRB).

### 2.1 Experiment 1: Optimization 1

**Participants** were 23 students (6 m, 17 f) aged 18 to 22 ($19.52 \pm 1.27$ yr).

**Stimuli** were a variety of 25 1D functions with different characteristics (linear/non-linear, differentiable/non-differentiable, etc.), including: *Polynomial, Exponential, Gaussian, Dirac, Sinc, etc.* The goal was to cover many cases and to investigate the generalization power of algorithms and humans.

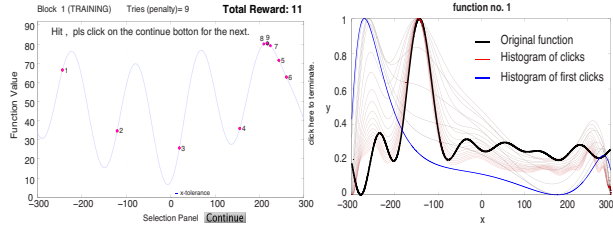

**Figure 1:** Left: a sample search trial. The unknown function (blue curve) was only displayed at the end of training trials. During search for the function's maximum, a red dot at $(x, f(x))$ was drawn for each $x$ selected by participants. Right: A sample function and the pdf of human clicks.

To generate a polynomial stimulus of degree $m$ ($m > 2$), we randomly generated $m + 1$ pairs of $(x, y)$ points and fitted a polynomial to them using least squares regression. Coefficients were saved for later use. Other functions were defined with pre-specified formulas and parameters (e.g., *Schwefel*, *Psi*). We generated two sets of stimuli, one for training and the other for testing. The $x$ range was fixed to $[-300\ 300]$ and the $y$ range varied depending on the function. Fig. 1 shows a sample search trial during training, as well as smoothed distribution of clicked locations for first clicks, and progressively for up to 15 clicks. In the majority of cases, the distribution of clicks starts with a strong leftward bias for the first clicks, then progressively focusing around the true function maximum as subjects make more clicks and approach the maximum. Subjects clicked less on smooth regions and more on spiky regions (near local maxima). This indicates that they sometimes followed the local gradient direction of the function.

**Procedure.** Subjects were informed about the goal of the experiment. They were asked to "find the maximum value (highest point) of a function in as few clicks as possible". During the experiment, each subject went through 30 test trials (in random order). Starting from a blank screen, subjects could click on any abscissa $x$ location, and we would show them the corresponding $f(x)$ ordinate. Previously clicked points remained on the screen until the end of the trial. Subjects were instructed to terminate the trial when they thought they had reached the maximum location within a margin of error ($\text{xTol}_h = 6$) shown at the bottom of the screen (small blue line in Fig. 1). This design was intentional to both obtain information about the human satisficing process and to make the comparison fair with algorithms (e.g., as opposed to automatically terminating a trial if humans happened to click near the maximum). We designed the following procedure to balance speed vs. accuracy. For each trial, a subject gained $A$ points for "HIT", lost $A$ points for "MISS", and lost 1 point for each click. Scores of subjects were kept on the record, to compete against other subjects. The subject with the highest score was rewarded with a prize. We used $A = 10$ (for 13 subjects) and $A = 20$ (for 10 subjects); since we did not observe a significant difference across both conditions, here we collapsed all the data. We highlighted to subjects that they should decide carefully where to click next, to minimize the number of clicks before hitting the maximum location. They were not allowed to click outside the function area. Before the experiment, we had a training session in which subjects completed 5 trials on a different set of functions than those used in the real experiment. The purpose was for subjects to understand the task and familiarize themselves to the general complexity and shapes of functions (i.e., developing a prior). We revealed the entire function at the end of each training trial only (not after test trials). The maximum number of clicks was set to 40. To prohibit subjects from using the vertical extent of the screen to guess the maximum location, we randomly elevated or lowered the function plot. We also recorded the time spent on each trial.

**Human Results.** On average, over all 25 functions and 23 subjects, subjects attempted $12.8 \pm 0.4$ tries to reach the target. Average hit rate (i.e., whether subjects found the maximum) over all trials was $0.74 \pm 0.04$. Across subjects, standard deviations of the number of tries and hit rates were $3.8$ and $0.74$. Relatively low values here suggest inter-subject consistency in our task.

Each trial lasted about $22 \pm 4$ seconds. Figure 2 shows example hard and easy stimuli ($fn$ are function numbers, see Supplement). The Dirac function had the most clicks (16.5), lowest hit rate (0.26), and longest time ($32.4 \pm 18.8$ s). Three other most difficult functions, in terms of function calls were, listed as (*function number, number of clicks*): $\{(f2,15.8), (f8,15.2), (f12,15.1)\}$. The easiest ones were: $\{(f16,9.3), (f20, 9.9), (f17, 10)\}$. For hit rate, the hardest functions were: $\{(f24,0.35), (f15,0.45), (f7,0.56)\}$, and the easiest ones: $\{(f20,1), (f17,1), (f22,0.95)\}$. Subjects were faster on *Gaussian (f17)* and *Exponential (f20)* functions (16.2 and 16.9 seconds).

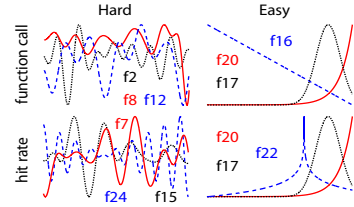

Figure 2: Difficult and easy stimuli.

**Model-based Results.** We compared human data to 24 well-established optimization algorithms. These baseline methods employ various search strategies (local, global, gradient-based, etc.) and often have several parameters (Table 1). Here, we emphasize one to two of the most important parameters for each algorithm. The following algorithms are considered: local search (e.g., Nelder-Mead simplex/FminSearch [10]); multi-start local search; population-based (e.g., Genetic Algorithms (GA) [12, 13], Particle Swarm Optimization (PSO) [11]); DIvided RECTangles (DIRECT) [15]; and Bayesian Optimization (BO) techniques [2, 3]. BO constructs a probabilistic model for $f(\cdot)$ using all previous observations and then exploits this model to make decisions about where along $\mathcal{X}$ to evaluate the function next. This results in a procedure that can find the maximum of difficult non-convex, non-differentiable functions with relatively few evaluations, at the cost of performing more computation to determine the next point to try. BO methods are based on Gaussian processes (GP) [9] and several selection criteria. Here we use a GP with a zero mean prior and a RBF covariance kernel. Two parameters of the kernel function, kernel width and signal variance, are learned from our training functions. We consider 5 types of selection criteria for BO: Maximum Mean (MM) [16], Maximum Variance (MV) [17], Maximum Probability of Improving (MPI) [18, 19], Maximum Expected Improvement (MEI)[20, 21], and Upper Confidence Bounds (UCB) [22]. Further, we consider two BO methods by Osborne *et al.* [23], with and without gradient information (See supplement). To measure to what degree human search behavior deviates from a random process, we devise a Random search algorithm which chooses the next point uniformly random from $[-300 \ 300]$ without replacement. We also run the Gradient Descent (GD) algorithm and its descendants denoted as minFunc-# in Table 1 where # refers to different methods (conjugate gradient (cg), quasi-Newton (qnewton), etc.).

Table 1: Baseline algorithms. We set maxItr to 500 when the only parameter is xTol$_m$.

| Algorithm | Type | Parameters |
|---|---|---|
| FminSearch [10] | Loc. | xTol$_m$ = 0.005:0.005:0.1 |
| FminBnd | L | xTol$_m$ = 5e-7:1e-6:1e-5 |
| FminUnc | L | xTol$_m$ = 0.01:0.05:1 |
| minFunc-# | L | xTol$_m$ = 0.01:0.05:1 |
| GD | L | xTol$_m$ = 0.001:0.001:0.005 |
| | | $\alpha$ = 0.1:0.1:0.5; tol = 1e-6 |
| mult-FminSearch | Glob. | xTol$_m$ = 0.005, starts = 1:10 |
| mult-FminBnd | G | xTol$_m$ = 5e-7, starts = 1:10 |
| mult-FminUnc | G | xTol$_m$ = 0.01, starts = 1:10 |
| PSO [11] | G | pop = 1:10; gen = 1:20 |
| GA [12, 13] | G | pop and gen = 5:10:100 |
| | | generation gap = 0.01 |
| SA [14] | G | stopTemp = 0.01:0.05:1 |
| | | $\beta$ = 0.1:0.1:1 |
| DIRECT [15] | G | maxItr = 5:5:70 |
| Random | G | maxItr = 5:5:150 |
| GP [2, 3] | G | maxItr = 5:5:35 |

To evaluate which algorithm better explains human 1D search behavior, we propose two measures: 1) an algorithm should have about the same performance, in terms of hit rate and function calls, as humans (1st-level analysis), and 2) it should have similar search statistics as humans, for example in terms of searched locations or search order (2nd-level analysis). For fair human-algorithm comparison, we simulate for algorithms the same conditions as in our behavioral experiment, when counting a trial as a hit or a miss (e.g., using same xTol$_h$). It is worth noting that in our behavioral experiment we did our best not to provide information to humans that we cannot provide to algorithms.

In the *1st-level analysis*, we tuned algorithms for their best accuracy by performing a grid search over their parameters to sample the hit-rate vs. function-calls plane. Table 1 shows two stopping conditions that are considered: 1) we either run an algorithm until a tolerance on $x$ is met (i.e., $|x_{i-1} - x_i| < \text{xTol}_m$), or 2) we allow it to run up to a variable (maximum) number of function calls (maxItr). For each parameter setting (e.g., a specified population size and generations in GA), since each run of an algorithm may result in a different answer, we run it 200 times to reach a reliable estimate of its performance. To generate a starting point for algorithms, we randomly sampled from

the distribution of human first clicks (over all subjects and functions, $p(x_1)$; see Fig. 1). As in the behavioral experiment, after termination of an algorithm, a hit is declared when: $\exists x_i \in B :$ $|x_i - \text{argmax}_x f(x)| \leq \text{xTol}_h$, where set $B$ includes the history of searched locations in a search trial. Fig. 3 shows search accuracy of optimization algorithms. As shown, humans are better than all algorithms tested, if hit rate and function calls are weighed equally (i.e., best is to approach the bottom-right corner of Fig. 3). That is, undergraduates from non-maths majors managed to beat the state of the art in numerical optimization. BO algorithms with GP-UCB and GP-MEI criteria are closer to human performance (so are GP-Osborne methods). The DIRECT method did very well and found the maximum with $\geq 30$ function calls. It can achieve better-than-human hit rate, with a number of function calls which is smaller than BO algorithms, though still higher than humans (it was not able to reach human performance with equal number of function calls). As expected, some algorithms reach human accuracy but with much higher number of function calls (e.g., GA, mult-start-#), sometimes by up to 3 orders of magnitude.

We chose the following promising algorithms for the *2nd-level analysis*: DIRECT, GP-Osborne-G, GP-Osborne, GP-MPI, GP-MUI, GP-MEI, GP-UCB, GP-UCB-Opt, GP-MV, PSO, and Random. GP-UCB-Opt is basically the same as GP-UCB with its exploration/exploitation parameter ($\kappa$ in $\mu_x + \kappa\sigma_x$; GP mean + GP std) learned from train data for each function. These algorithms were chosen because their performance curve in the first analysis intersected a window where accuracy is half of humans and function call is twice as humans (black rectangle in Fig. 3). We first find those parameters that led these algorithms to their closest performance to humans. We then run them again and this time save their searched locations for further analysis.

We design 4 evaluation scores to quantify similarities between algorithms and humans on each function: 1) mean sequence distance between an algorithm's searched locations and human searched locations, in each trial for the first 5 clicks, 2) mean shortest distance between an algorithm's searched locations and all human clicks (i.e., point matching), 3) agreement between probability distributions of searched locations by all humans and an algorithm, and 4) agreement between pdfs of normalized step sizes (to [0 1] on each trial). Let $p_m(t)$ and $p_h(t)$ be pdfs of the search statistic $t$ by an algorithm and humans, respectively. The agreement between $p_m$ and $p_h$ is defined as $p_m(\text{argmax}_t \, p_h(t))$ (i.e., the value of an algorithm's pdf at the location of maximum for human pdf). Median scores (over all 25 functions) are depicted in Fig. 4. Dis-

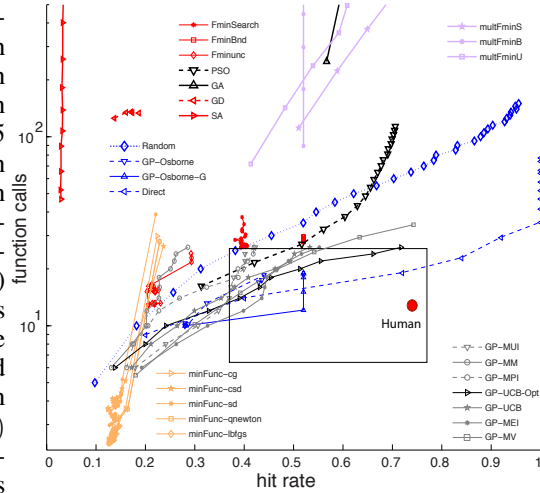

Figure 3: Human vs. algorithm 1D search accuracy.

tance score is lower for Bayesian models compared to DIRECT, Random, PSO, and GP-Osborne algorithms (Fig. 4.a). Point matching distances are lower for GP-MPI, and GP-UCB (Fig. 4.b). These two algorithms also show higher agreement to humans in terms of searched locations (Fig. 4.c). The clearest pattern happens over step size agreement with BO methods (except GP-MV) being closest to humans (Fig. 4.d). GP-MPI and GP-UCB show higher resemblance to human search behavior over all scores. Further, we measure the *regret* of algorithms and humans defined as $f_{\text{max}}(\cdot) - f^*$ where $f^*$ is the best value found so far for up to 15 function calls averaged over all trials. As shown in Fig. 4.e, BO models approach the maximum of $f(\cdot)$ as fast as humans. Hence, although imperfect, BO algorithms overall are the most similar to humans, out of all algorithms tested.

Three reasons prompt us to consider BO methods as promising candidates for modeling human basic search: 1) BO methods perform efficient search in a way that resembles human behavior in terms of accuracy and search statistics (results of Exp. 1), 2) BO methods exploit GP which offers a principled and elegant approach for adding structure to Bayesian models (in contrast to purely data-driven Bayesian). Furthermore, the sequential nature of the BO and updating the GP posterior after each function call seems a natural strategy humans might be employing, and 3) GP models explain function learning in humans over simple functional relationships (linear and quadratic) [24]. Function learning and search mechanisms are linked in the sense that, to conduct efficient search, one needs to know the search landscape and to progressively update one's knowledge about it.

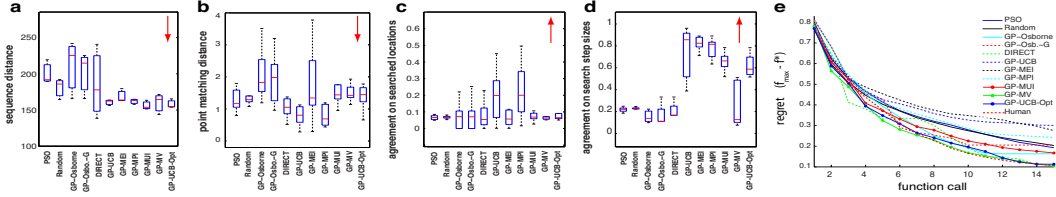

Figure 4: Results of our second-level analysis. The lower the distance and the higher the agreement, the better (red arrows). Boxes represent median (red line) and $25\,th$, $75\,th$ percentiles. Panel **(e)** shows average regret of algorithms and humans ($f^*$ is normalized to $f_{\max}$ for each trial separately).

We thus designed 6 additional controlled experiments to further explore GP as a unified computational principle guiding human function learning and active search. In particular, we investigate the basic idea that humans might be following a GP, at least in the continuous domain, and change some of its properties to cope with different tasks. For example, humans may use GP to choose the next point to dynamically balance exploration vs. exploitation (e.g., in search task), or to estimate the function value of a point (e.g., in function interpolation). In experiments 2 to 5, subjects performed interpolation and extrapolation tasks, as well as active versions of these tasks by choosing points to help them learn about the functions. In experiments 6 and 7, we then return to the optimization task, for a detailed model-based analysis of human search behavior over functions from the same family. Note that many real-world problems can be translated into our synthetic tasks here.

We used polynomial functions of degree 2, 3, and 5 as our stimuli (denoted as Deg2, Deg3, and Deg5, respectively). Two different sets of functions were generated for training and testing, shown in Fig. 5. For each function type, subjects completed 10 training trials followed by 30 testing trials. As in Exp. 1, function plots were disclosed to subjects only during training. To keep subjects engaged, in addition to the competition for a prize, we showed them the magnitude of error during both training and testing sessions. In experiments 2 to 6, we fitted a GP to different types of functions using the same set of $(x, y)$ points shown to subjects during training (Fig. 6). A grid search was conducted to learn GP parameters from the training functions to predict subjects' test data.

## 2.2 Experiments 2 & 3: Interpolation and Active Interpolation

**Participants.** Twenty subjects (7m, 13f) aged 18 to 22 participated (mean: $19.75 \pm 1.06$ yr).

**Procedure.** In the interpolation task, on each function, subjects were shown 4 points $x \in \{-300, a, b, 300\}$ along with their $f(x)$ values. Points $a$ and $b$ were generated randomly once in advance and were then tied to each function. Subjects were asked to guess the function value at the center ($x = 0$) as accurately as possible. In the active interpolation task, the same 4 points as in interpolation were shown to subjects. Subjects were first asked to choose a 5th point between $[-300\ 300]$ to see its $y = f(x)$ value. They were then asked to guess the function value at a randomly-chosen 6th $x$ location as accurately as possible. Subjects were instructed to pick the most informative fifth point regarding estimating the function value at the follow-up random $x$ (See Fig. 6).

**Results.** Fig. 7.a shows mean distance of human clicks from the GP mean at $x = 0$ over test trials (averaged over absolute pairwise distances between clicks and the GP) in the interpolation task. Human errors rise as functions become more complicated. Distances of the GP and the actual function from humans are the same over Deg2 and Deg3 functions (no significant difference in medians, Wilcoxon signed-rank test, $p > 0.05$). Interestingly, on Deg5 functions, GP is closer to human clicks than the actual function (signed-rank test, $p = 0.053$) implying that GP captures clicks well in this case. GP did fit the human data even better than the actual function, thereby lending support to our hypothesis that GP may be a reasonable approximation to human function estimation.

Could it be that subjects locally fit a line to the two middle points to guess $f(0)$? To evaluate this hypothesis, we measured the distance, at $x = 0$, from human clicks to a line passing through $(a, f(a))$ and $(b, f(b))$. By construction of our stimuli, a line model explains human data well on Deg3 and Deg5, but fails dramatically on Deg2 which deflect around the center. GP is significantly better than the line model on Deg2 ($p < 0.0005$), while being as good on Deg3 and Deg5 ($p = 0.78$).

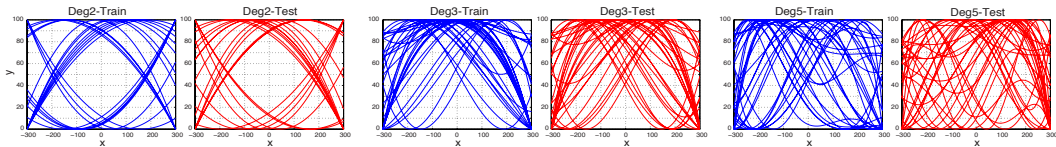

Figure 5: Train and test polynomial stimuli used in experiments 2 through 6.

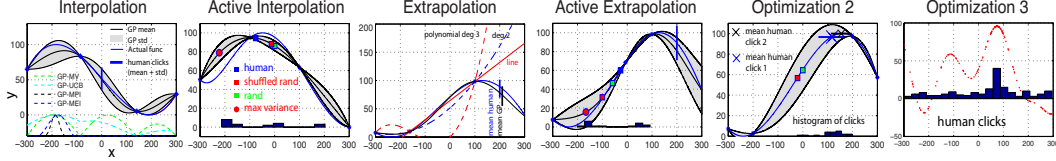

Figure 6: Illustration of experiments. In extrapolation, polynomials (degrees 1, 2 & 3) fail to explain our data.

Another possibility could be that humans choose a point randomly on the $y$ axis, thus discarding the shape of the function. To control for this, we devised two random selection strategies. The first one uniformly chooses $y$ values between 0 and 100. The second one, known as *shuffled random (SRand)*, takes samples from the distribution of $y$ values selected by other subjects over all functions. The purpose is to account for possible systematic biases in human selections. We then calculate the average of the pairwise distances between human clicks and 200 draws from each random model. Both random models fail to predict human answers on all types of functions (significantly worse than GP, signed-rank test $ps < 0.05$). One advantage of the GP over other models is providing a level of uncertainty at every $x$. Fig. 7.a (inset) demonstrates similar uncertainty patterns for humans and the GP, showing that both uncertainties (at $x = 0$) rise as functions become more complicated.

Interpolation results suggest that humans try to capture the shape of functions. If this is correct, we expect that humans will tend to click on high uncertainty regions (according to GP std) in the active interpolation task (see Fig. 6 for an example). Fig. 7.b shows the average of GP standard deviation at locations of human selections. Humans did not always choose $x$ locations with the highest uncertainty (shown in red in Fig. 7.b). One reason for this might be that several regions had about the same std. Another possibility is because subjects had slight preference to click toward center. However, GP std at human-selected locations was significantly higher than the GP std at random and SRand points, over all types of functions (signed-rank test, $ps < 1e{-}4$; non-significant on Deg2 vs. SRand $p = 0.18$). This result suggests that since humans did not know in advance where a follow-up query might happen, they chose high-uncertainty locations according to GP, as clicking at those locations would most shrink the overall uncertainty about the function.

## 2.3 Experiments 4 & 5: Extrapolation and Active Extrapolation

**Participants.** 16 new subjects (7m, 9f) completed experiments 4 and 5 (Age: $19.62 \pm 1.45$ yr).
**Procedure.** Three points $x \in \{-300, c, 100\}$ and their $y$ values were shown to subjects. Point $c$ was random, specific to each function. In the extrapolation task, subjects were asked to guess the function value at $x = 200$ as accurately as possible (Fig. 6). In the active extrapolation task, subjects were asked to choose the most informative 4th point in $[-300\ 100]$ regarding estimating $f(200)$.

**Results.** A similar analysis as in the interpolation task is conducted. As seen in Fig. 7.c, in alignment with interpolation results, humans are good at Deg2 and Deg3 but fail on Deg5, and so does the GP model. Here again, with Deg5, GP and humans are closer to each other than to the actual function, further suggesting that their behaviors and errors are similar. There is no significant difference between GP and the actual functions over all three function types (signed-rank test; $p > 0.25$). Interestingly, a line model fitted to points $c$ and 100 is impaired significantly ($p < 1e{-}5$ vs. GP) over all function types (Fig. 6). Both random strategies also performed significantly worse than GP on this task (signed-rank test; $ps < 1e{-}6$). SRand performs better than uniform random, indicating

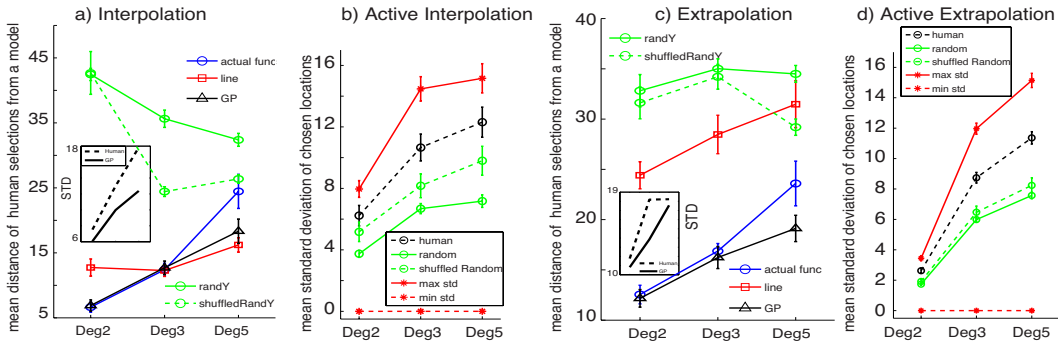

Figure 7: **a)** Mean distance of human clicks from models. Errors bars show standard error of the mean (s.e.m) over test trials. Inset shows the standard deviation of humans and the GP model at $x = 0$. **b)** mean GP std at human vs. random clicks in active interpolation. **c** and **d** correspond to **a** and **b**, for the extrapolation task.

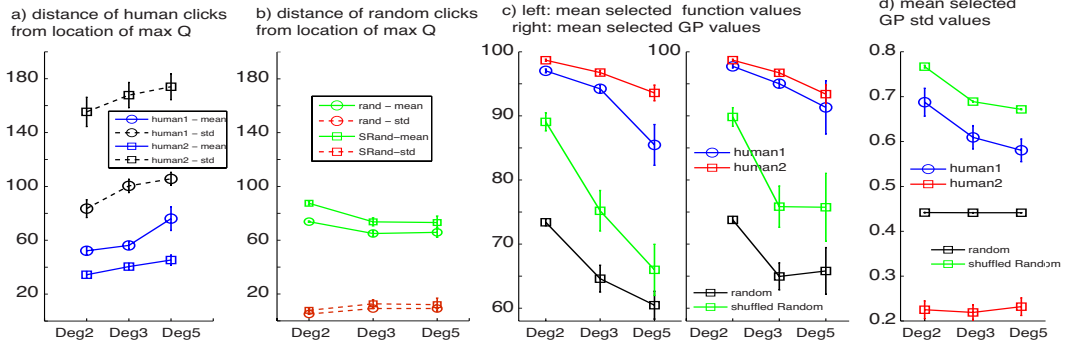

Figure 8: Results of the optimization task 2. **a** and **b**) distance of human and random clicks from locations of max Q (i.e., GP mean and max GP std). **c**) actual function and GP mean values at human and random clicks. **d**) normalized GP standard deviation at human vs. random clicks. Errors bars show s.e.m over test trials.

existence of systematic biases in human clicks. Subjects learned that $f(200)$ did not happen on extreme lows or highs (same argument is true for $f(0)$ in interpolation). As in the interpolation task, GP and human standard deviations rise as functions become more complex (Fig. 7.c; inset). Active extrapolation (Fig. 7.d), similar to active interpolation, shows that humans tended to choose locations with significantly higher uncertainty than uniform and SRand points, for all function types ($ps < 0.005$). Some subjects in this task tended to click toward the right (close to 100), maybe to obtain a better idea of the curvature between 100 and 200. This is perhaps why the ratio of human std to max std is lower in active extrapolation compared to active interpolation (0.75 vs. 0.82), suggesting that maybe humans used an even more sophisticated strategy on this task.

## 2.4 Experiment 6: Optimization 2

**Participants** were another 21 subjects (4m, 17f) in the age range of 18 to 22 (mean: $20 \pm 1.18$).
**Procedure.** Subjects were shown function values at $x \in \{-300, -200, 200, 300\}$ and were asked to find the $x$ location where they think the function's maximum is. They were allowed to make two equally important clicks and were shown the function value after each one. For quadratic functions, we only used 13 concave-down functions that have one unique maximum.

**Results.** We perform two analyses shown in Fig. 8. In the first one, we measure the mean distance of human clicks (1st and 2nd clicks) from the location of the maximum GP mean and maximum GP standard deviation (Fig. 8.a). We updated the GP after the first click.

We hypothesized that the human first click would be at a location of high GP variance (to reduce uncertainty about the function), while the second click would be close to the location of highest GP mean (estimated function maximum). However, results showed that human 1st clicks were close to the max GP mean and not very close to the max GP std. Human 2nd clicks were even closer (signed-rank test, $p < 0.001$) to the max GP mean and further away from the max GP std ($p < 0.001$). These two observations together suggest that humans might have been following a Gaussian process with a selection criterion heavily biased towards finding the maximum, as opposed to shrinking the most uncertain region. Repeating this analysis for random clicks (uniform and SRand) shows quite the opposite trend (Fig. 8.b). Random locations are further apart from maximum of the GP mean (compared to human clicks) while being closer to the maximum of the GP std point (compared to human clicks). This cross pattern between human and random clicks (wrt. GP mean and GP std) shows a systematic search strategy utilized by humans. Distances of human clicks from the max GP mean and max GP std rise as functions become more complicated. In the second analysis (Fig. 8.c), we measure actual function and GP values at human and random clicks. Humans had significantly higher function values at their 2nd clicks; $p < 1e$–4 (so was true using GP; $p < 0.05$). Values at random points are significantly lower than human clicks. Humans were less accurate as functions became more complex, as indicated by lower function values. Finally, Fig. 8.d shows that humans chose points with significantly less std (normalized to the entire function) in their 2nd clicks compared to random and first clicks. Human 1st clicks have higher std than uniform random clicks.

## 2.5 Experiment 7: Optimization 3

**Participants** were 19 new subjects (6m, 13f) in the age range of 19 to 25 (mean: $20.26 \pm 1.64$ yr).
**Stimuli.** Functions were sampled from a Gaussian process with predetermined parameters to assure functions come from the same family and resemble each other (as opposed to Exp. 1; See Fig. 6).
**Procedure** was the same as in Exp. 1. Number of train and test trials, in order, were 10 and 20.

**Results.** Subjects had average accuracy of $0.76 \pm 0.11$ ($0.5 \pm 0.18$ on train) over all subjects and functions, and average clicks of $8.86 \pm 1.12$ ($7.15 \pm 1.2$ on train) before ending a search trial. To investigate the sequential strategy of subjects, we progressively updated a GP using a subject's clicks on each trial, and exploited this GP to evaluate the next click of the same subject. In other words, we attempted to know to what degree a subject follows a GP. Results are shown in Fig. 9. The regret of the GP model and humans decline with more clicks, implying that humans chose informative clicks regarding optimization (figure inset). Humans converge to the maximum location slightly faster than a GP fitted to their data, and much faster than random.

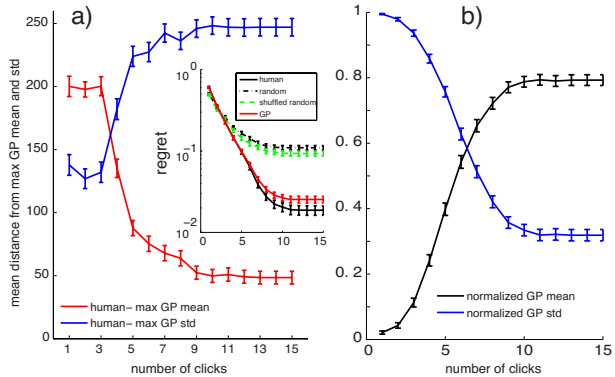

Figure 9: Exploration vs. exploitation balance in Optimization 3 task.

Fig. 9.a shows that subjects get closer to the location of maximum GP mean and further away from max GP std (for 15 clicks). Fig. 9.b shows the normalized mean and standard deviation of human clicks (from the GP model), averaged over all trials. At about 6.4 clicks, subjects are at 58% of the function maximum while they have reduced the variance by 42%. Interestingly, we observe that humans tended to click on higher uncertainty regions (according to GP) in their first 6 clicks (average over all subjects and functions), then gradually relying more on the GP mean (i.e., balancing exploration vs. exploitation). Results of optimization tasks suggest that human clicks during search for a maximum of a 1D function can be predicted by a Gaussian process model.

## 3 Discussion and Conclusion

Our contributions are twofold: First, we found a striking capability of humans in 1D function optimization. In spite of the relative naivety of our subjects (not maths or engineering majors), the high human efficiency in our search task does open the challenge that even more efficient optimization algorithms must be possible. Additional pilot investigations not shown here suggest that humans may perform even better in optimization when provided with first and second derivatives. Following this road may lead to designing efficient selection criteria for BO methods (for example new ways to augment gradient information with BO). However, it remains to be addressed how our findings scale up to higher dimensions and benchmark optimization problems. Second, we showed that Gaussian processes provide a reasonable (though not perfect) unifying theoretical account of human function learning, active learning, and search (GP plus a selection strategy). Results of experiments 2 to 5 lead to an interesting conclusion: In interpolation and extrapolation tasks, subjects try to minimize the error between their estimation and the actual function, while in active tasks they change their objective function to explore uncertain regions. In the optimization task, subjects progressively sample the function, update their belief and use this belief again to find the location of maximum. (i.e., exploring new parts of the search space and exploiting parts that look promising).

Our findings support previous work by Griffiths *et al.* [24] (also [25, 26, 27]). Yet, while they showed that Gaussian processes can predict human errors and difficulty in function learning, here we focused on explaining human active behavior with GP, thus extending explanatory power of GP one step ahead. One study showed promising evidence that our results may extend to a larger class of natural tasks. Najemnik and Geisler [6, 28, 29] proposed a Bayesian ideal-observer model to explain human eye movement strategies during visual search for a small Gabor patch hidden in noise. Their model computes posterior probabilities and integrates information across fixations optimally. This process can be formulated with BO with an exploitative search mechanism (i.e., GP-MM). Castro *et al.* [30] studied human active learning on the well-understood problem of finding the threshold in a binary search problem. They showed that humans perform better when they can actively select samples, and their performance is nearly optimal (below the theoretical upper bound). However, they did not address how humans choose the next best sample. One aspect of our study which we did not elaborate much here, is the satisficing mechanisms humans used in our task to decide when to end a trial. Further modeling of our data may be helpful to develop new stopping criteria for active learning methods. Related efforts have studied strategies that humans may use to quickly find an object (i.e., search, active vision) [31, 32, 33, 34, 35], optimal foraging [36], and optimal search theories [4, 5], which we believe could all now be revisited with GP as an underlying mechanism.

Supported by NSF (CCF-1317433, CMMI-1235539) and ARO (W911NF-11-1-0046, W911NF-12-1-0433).

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
