[Supplementary Material]

# Supplement to
# "Bayesian optimization explains human active search"
# NIPS 2013

Ali Borji and Laurent Itti
{borji,itti}@usc.edu

November 7, 2013

## 1  Stimuli in Optimization Task 1

Figure 1: Illustration of stimuli used during test trials. Functions are reconstructed from human clicked locations. F18 is the *Dirac* function with a non-zero element at -270.

Figure 2: Progressive distribution of human clicks in optimization task 1 progressively up to 15 clicks (corresponding to Fig. 1 in main text).

Figure 3: Distribution of first clicks over all subjects and functions. This pdf is used to draw the initial $x$ for optimization algorithms.

Table 1: Equations of stimuli functions.

| No. | Functions |
|-----|-----------|
| 1 | $f(x) = -4.6e\text{-}6\ x^4 - 1.05e\text{-}4\ x^3 + 0.03\ x^2 + 0.39\ x + 7.9$ |
| 2 | $f(x) = 6.3\ e\text{-}6\ x^4 + 5.89\ e\text{-}4 x^3 - 0.012 x^2 - 1.61 x + 28.61$ |
| 3 | $f(x) = -6.1e\text{-}7\ x^5 - 9e\text{-}5\ x^4 - 7.2e\text{-}4\ x^3 + .3 x^2 + 11.8 x + 1.46$ |
| 4 | $f(x) = -9.18e\text{-}9 x^5 + 2.77e\text{-}6\ x^4 - 0.01 x^2 - 0.69 x + 65.9$ |
| 5 | $f(x) = 2.34e\text{-}5 x^4 + 8.2e\text{-}5\ x^3 - 0.03 x^2 - 0.21 x + 56.8$ |
| 6 | $f(x) = 5.4e\text{-}8 x^5 + 4.5e\text{-}6 x^4 - 0.02 x^2 + 0.86 x + 62.6$ |
| 7 | $f(x) = 1.79e\text{-}7 x^4 - 0.04 x + 58.4$ |
| 8 | $f(x) = 3.39e\text{-}6 x^4 + 0.77 x + 41.7$ |
| 9 | $f(x) = 2.9e\text{-}6 x^6 - 6.2e\text{-}5 x^3 - 0.01 x^3 - 0.07 x^2 + 9.4 x + 195.4$ |
| 10 | $f(x) = 3.5e\text{-}7 x^6 + 5.3e\text{-}6 x^5 - 0.01 x^3 + 0.13 x^2 + 4.4 x + 39.3$ |
| 11 | $f(x) = -3.6e\text{-}8 x^5 - 1.4e\text{-}5 x^4 + 0.03 x^2 - 0.22 x + 33.2$ |
| 12 | $f(x) = 1.1e\text{-}7\ x^5 + 1.08e\text{-}5 x^4 + 7.2e\text{-}5 x^3 - 0.01 x^2 - 0.28 x + 87$ |
| 13 | $f(x) = 1.5e\text{-}5 x^4 - 9.122e\text{-}5 x^3 - 0.04 x^2 + 0.69 x + 81.9$ |
| 14 | $f(x) = 6.49e\text{-}8 x^5 + 9.99e\text{-}6 x^4 - 0.02 x^2 + 0.40 x + 53.9$ |
| 15 | $f(x) = -1.06e\text{-}7 x^4 - 1.98e\text{-}5 x^3 + 1.10 x^2 + 18 x + 83.1$ |
| 16 | $f(x) = 7\texttt{schw}(x - 50);\ \texttt{schw}(x) = 837.96 - x sin(\sqrt{|x|})$ |
| 17 | $f(x) = -psi(|100(x - 100)|);\ f(100) = -4.6$ |
| 18 | $f(x) = 1$  for $x = -270$; otherwise $f(x) = 0$ |
| 19 | $f(x) = \frac{1}{\sigma\sqrt{2\pi}} e^{\frac{-(x-100-\mu)^2}{2\sigma^2}};\ \mu = 100,\ \sigma = 50$ |
| 20 | $f(x) = (x + 50) \times cos(0.03\pi x)$ |
| 21 | $f(x) = sinc(0.03(x - 200))$ |
| 22 | $f(x) = -(2x)^3 + 5000$ |
| 23 | $f(x) = -(x + 100)^2$ |
| 24 | $f(x) = e^{0.02(x-100)}$ |
| 25 | $f(x) = -x + 100$ |

# 2 Baseline Optimization Algorithms

Some algorithms perform local search including: fminSearch, fminBnd, and fminUnc. fminSearch algorithm (also known as Nelder-Mead simplex method [1]) finds the minimum of a scalar function, starting at an initial estimate. This is a direct search method for unconstrained nonlinear optimization and does not use numerical or analytic gradients. fminBnd algorithm finds minimum of a single-variable function on a fixed interval. It uses the golden section search and parabolic interpolation. fminUnc algorithm uses the BFGS Quasi-Newton method with a cubic line search procedure. We used Matlab for running these algorithms.

We also run the Gradient Descent (GD) algorithm and its descendants using the minFunc-# code[1] where # refers to different Gradient approaches (e.g., Conjugate Gradient (cg), Quasi-Newton (qnewton), etc.). Each iteration in these algorithms involves calculating a *search direction*, and then querying function values along that direction until certain conditions are met. In GD, parameters $\alpha$ and *tol* control step size and tolerance on gradient norm, respectively.

To assess whether humans randomly pick a point and then follow the gradient, we run three multi-start local search algorithms using fminSearch, fminBnd, and fminUnc. For these algorithms, in addition to $\text{xTol}_m$, we also change the number of random starts.

Some algorithms are inspired by natural phenomena and are population-based. They are derivative-free and are suitable for scenarios where it is not possible to calculate the derivative of the objective function. We explore two types: Particle Swarm Optimization (PSO)[2] [2] and Genetic Algorithms (GA) [3, 4][3]. For GA, we use the stochastic universal sampling, single-point crossover (rate = 0.7), and bit flip mutation (rate = 0.06). For PSO, we set cognitive and social attraction parameters to 0.5 and 0.1, respectively.

Simulated Annealing (SA) [5] models the physical process of heating a material and then slowly lowering the temperature to decrease defects, thus minimizing the system energy while avoiding local extrema (if the temperature is lowered with a sufficiently slow schedule). Here we vary the stopping temperature and cooling schedule ($\beta$).

DIRECT [6] is a derivative-free sampling algorithm. Similar to BO approaches, it samples points in the domain, and uses the information it has obtained to decide where to search next. DIRECT can be very useful when the objective function is a "black box" function or simulation. It has been shown to be very competitive with existing algorithms, with the advantage of requiring few parameters [7].

Note that our main goal here is to discover which algorithm explains human search behavior better. See [8] for benchmarking of optimization algorithms.

# 3 Bayesian Global Optimization

## 3.1 Gaussian Processes

GP is used to probabilistically model an unknown function based on prior information which includes the set of existing observed samples and their obtained function values $\mathcal{I} = \{(x_1, y_1), (x_2, y_2), \cdots, (x_n, y_n)\}$ where $y_i = f(x_i)$. Let $\mathcal{I} = \{\mathbf{x}_\mathcal{I}, \mathbf{y}_\mathcal{I}\}$ where $\mathbf{x}_\mathcal{I} = \{x_1, x_2, \cdots, x_n\}$ and $\mathbf{y}_\mathcal{I} = \{y_1, y_2, \cdots, y_n\}$ represent the prior information. The goal for the posterior model is to predict the function output for any $x \in \mathcal{X}^d \backslash \mathbf{x}_\mathcal{I}$. For any unobserved point, the Gaussian process models its function output as a normal random variable, with its mean predicting the expected function output of the point, and the variance indicating the uncertainty associated with the prediction. GP has a convenient closed form for estimating the conditional posterior mean and variance of the function value $y$:

$$
\begin{aligned}
y|\mathcal{I} &\sim \mathcal{N}(\mu, \sigma^2) \\
\mu &= k(x, \mathbf{x}_\mathcal{I}) k(\mathbf{x}_\mathcal{I}, \mathbf{x}_\mathcal{I})^{-1} \mathbf{y}_\mathcal{I} \\
\sigma^2 &= k(x, x) - [k(x, \mathbf{x}_\mathcal{I}) k(\mathbf{x}_\mathcal{I}, \mathbf{x}_\mathcal{I})^{-1} k(\mathbf{x}_\mathcal{I}, x)]
\end{aligned}
\tag{1}
$$

where $k(.,.)$ is an arbitrary symmetric positive definite kernel function that specifies the element of the covariance matrix. Here, we use a zero mean GP prior and covariance specified by a RBF kernel function (known as *Squared Exponential*): $k(x_i, x_j) = \sigma_f exp(-\frac{1}{l}|x_i - x_j|^2)$ where $l$ is the kernel width (scale parameter) that can be considered as the distance we need to move in the input space before the function value changes significantly. $\sigma_f$ is the signal variance which specifies the maximum possible variance at each point. We conducted a grid search to learn $l$ and $\sigma_f$ from training functions. The resultant parameters $l = 500$ and $\sigma_f = 100$ were used for Bayesian optimization models. We also empirically verified that the GP behaved reasonably well on our functions.

For interpolation and extrapolation tasks, GP parameters were learned from human data (i.e., human responses over train trails and not the actual functions as opposed to Experiment 1). The learned GP parameters were then used to predict human answers over test trials.

## 3.2 Selection Criteria

Below we explain the 5 most popular sequential selection criteria in the literature, which we use here. Note that selection criteria are not limited to these and additional ones have recently been proposed (e.g., Information theoretic BO [9, 10], GP-Hedge [11], Thompson sampling [12]).

**Maximum Mean (MM):** This purely exploitative (greedy) heuristic (a.k.a. PMAX) selects the point which has the highest output mean [13, 14]: $x_{\mathtt{next}} = \mathtt{argmax}_x \mu$.

**Maximum Variance (MV):** This purely exploratory heuristic (a.k.a. uncertainty sampling [15]) selects the point which has the highest output variance (or standard deviation): $x_{\mathtt{next}} = \mathtt{argmax}_x \sigma$.

**Maximum Probability of Improving (MPI):** One intuitive strategy is to maximize the probability of improving over the current best observation, $y_{max}$, by a given margin $\alpha$ [16, 17, 18]. The next sample is the one that will produce an output no smaller than $(1 + \alpha)y_{max}$ with the

highest probability: $x_{\texttt{next}} = \texttt{argmax}_x \; p(y(x) \geq (1 + \alpha)y_{max})$. Using GP properties we can write:

$$x_{\texttt{next}} = \texttt{argmax}_x \; \Phi\left(\frac{\mu_x - (1 + \alpha)y_{max}}{\sigma_x}\right) \tag{2}$$

$\Phi(.)$ is the normal cumulative distribution function.

**Maximum Expected Improvement (`MEI`):** This criterion selects the sample that directly maximizes the *expected* improvement [19, 20]. It can be written as the following closed form:

$$x_{\texttt{next}} = \texttt{argmax}_x \; \sigma_x\big[-u\Phi(-u) + \phi(u)\big] \tag{3}$$

where $u = \frac{y_{max} - \mu_x}{\sigma_x}$ and $\phi(.)$ is the normal pdf.

**Upper Confidence Bounds (`UCB`):** To overcome the greedy nature of `MM`, [21, 22] proposed the Sequential Design for Optimization (`SDO`) algorithm, in which they exploited upper confidence bounds (lower, when considering minimization) to construct acquisition functions of the form:

$$x_{\texttt{next}} = \texttt{argmax}_x \; \texttt{UCB}(x) = \texttt{argmax}_x \; \mu_x + \kappa\sigma_x \tag{4}$$

where parameter $\kappa$ balances exploitation against exploration. Here, we set $\kappa$ to 5 to weigh variance more than mean, we eventually tuned this parameter to achieve its best accuracy for each function (denoted as GP-UCB-Opt in the paper). A similar version of this heuristic (posed as a multi-armed bandit problem), called Maximum Upper-bound Interval (`MUI`) [23], explores areas with non-trivial probability of achieving a good result, as measured by the upper bound of the 95% confidence interval of output prediction: $x_{\texttt{next}} = \texttt{argmax}_x \; \mu_x + 1.96\sigma_x$.

In addition to the above, we run two recent BO methods proposed by [24] referred here as GP-Osborne and GP-Osborne-G. The latter also uses the local gradient information.

# 4 Convergence of Hit Rate and Function Calls

Figure 4: Convergence of the fminSearch (running average of hit rate and function calls) over 20 executions for two functions. Similar patterns happen over other algorithms.

# 5 Stimuli in Interpolation and Extrapolation Tasks

Polynomials of degree 2; Deg2:

$$y = p_1 - p_2 \times (x - p_3)^2 \tag{5}$$

Polynomials of degree 3; Deg3:

$$y = (x - p_1) \times (x - p_2) \times (x - p_3) \times p_4 + p_5 \tag{6}$$

Polynomials of degree 5; Deg5:

$$y = (x - p_1) \times (x - p_2) \times (x - p_3) \times (x - p_4) \times (x - p_5) \times p_6 + p_7 \tag{7}$$

We randomly generated two sets of coefficients for training and testing trials shown in Fig. 6 in the main text. Functions are then normalized to the range of [0 100].

# 6 Stimuli in Optimization Task 3

Figure 5: Test stimuli used in optimization task 3. Functions are sampled from a Gaussian process with known parameters.

## Footnotes

[1] http://www.di.ens.fr/~mschmidt/Software/minFunc

[2] http://code.google.com/p/psomatlab/.

[3] We use **gatbx** toolbox at: http://www.geatbx.com/