[Reviews · NeurIPS 2013]

Submitted by Assigned_Reviewer_7

In this well-written and dense paper the authors compare human active learning, optimization, and inter/extrapolation of 1D functions. Their two conclusions are that (a) people outperform modern optimization algorithms, and (b) their behavior is largely consistent with bayesian inference/search/prediction based on gaussian process assumptions about function smoothness.

I have some concerns about point (a) (arguably less important than point b). I'm not convinced that the authors eliminated all the additional information people have. Critically, the range of the y axis remains an issue. The authors say that they jittered the y offset to alleviate the problem, but I think numbers here are important: what is the range of the function, relative to the displayed range (the remainder being the range of the jitter). Are these relative ranges constant across trials? In general, if a point appears at the bottom of the screen, subjects know that it must be far from the maximum, if it appears close to the top, it must be close to the maximum y value. None of the optimization algorithms know this, and thus can't use it to help search.

While this paper has a lot of interesting measures, which each speak to different aspects of the (mis)match between algorithm and human behavior, I find this array of measures confusing, and largely unnecessary. How about a single measure as follows:
give the algorithm the first K points a subject observed. Have the algorithm pick point K+1. Compare the K+1th point for the algorithm and for the person.
Maybe something else would work better. However, looking at figure 4, I am overwhelmed by the noisiness and apparent lack of diagnosticity of the assorted measures. I also find it hard to assess why agreement on the distribution of search step sizes is important.

Human consistency. Instead of the convoluted array of measures, I'd like to see some space dedicated to assessing across-subject agreement. The low standard deviations of crude performance across people are suggestive of consistency, but not enough. How high are the split-half correlations of performance over the different trials? How well does the distribution of clicks from one half of the subjects predict the distribution of clicks from the other half? This later measure I find particularly important to be able to compare the across-subject agreement to the agreement with various algorithms (this could be done for most of the marginal measures the authors use).

I'm not sure why in Expt 6 the authors thought the first click should be to the maximum variance point. I think a complete decision theoretic analysis of the problem is important. Given that both clicks are "equally important" (ok, admittedly I don't know what the authors meant here), it's not so clear that the correct strategy is to gain maximal information on the first click. if the maximum is unambiguous, wouldn't that be the correct place to click?

Minor points: I was a bit confused about the claim that subjects use the 'gradient': of course, they don't have access to the gradient, so they use something like the 'estimated smoothness' -- which is another further point to argue for some gaussian process like algorithm that does this sort of estimation.
Summary: The paper has a rich set of (somewhat redundant) experiments on human optimization assessed via a large model bake-off. It would be improved by using fewer, more thoughtful measures by which to evaluate model fit and by more careful estimates of across-subject consistency.

Submitted by Assigned_Reviewer_8

Paper 68 – Bayesian optimization explains human active search

The authors explore different optimization strategies for 1-D continuous functions and their relationship to how people optimize the functions. They used a wide variety of continuous functions (with one exception): polynomial, exponential, trigonometric, and the Dirac function. They also explore how people interpolate and extrapolate noisy samples from a latent function (which has a long tradition in psychology under the name of function learning) and how people select an additional sample to observe under the task of interpolating or extrapolating. Over all, they found that Gaussian processes do a better job at describing human performance than any of the approx. 20 other tested optimization methods.

Broadly, I really enjoyed this paper and I believe that it is a strong NIPS submission. They tackled interesting problems for machine learning-oriented cognitive scientists (although the stimuli may be a bit too abstract to be of interest for many cognitive psychologists): optimizing a function, and active interpolation and extrapolation. It is well written and besides for some minor quibbles, the presentation is very good.

Besides for some minor points that I enumerate below, I have two main criticisms of the paper.

1. In contrast to most function learning experiments, the experimental stimuli and procedure were maximally abstract (dots from a function and “find the maximum value of a function”). This is not ecologically valid and thus, it is difficult to interpret what the mean for actual human function learning. The functions used in the experiments are probably not those functions that people expect in their everyday lives. Additionally, there typically is a strong effect of context on this sort of human learning and it is not clear how their results can be extended to account for context (which in my opinion would be a very interesting direction for them to go into for future research.). It is not enough to merely state “note that many real-world problems can be translated into our synthetic tasks here.” There needs to be examples and a justification for why the synthetic tasks can be generalized to real-world examples. However, they got compelling results, where participants seem to be doing the task.

2. I would have liked to see the authors integrate their experiments and results better with previous human function learning work. For example, I do not see why they did not choose to do their experiment using the standard cover story and stimuli of a function learning experiment. I understand why the authors did not just use a positive or negative linear function (although given f20 of experiment 1 is unbounded, it seems reasonable to include these too), but it seems including a piecewise linear function would have been a smart choice. As f16 is nearly a negative linear function and has been previously shown to be easy to learn, it is sensible that it would be one of the easiest function to maximize (and this is the sort of integration with previous work that I would have liked to have seen). Integrating with previous work would have made the paper stronger and the results more interpretable.

More minor concerns:

Personally, I would have preferred to see the previous psychological work to be earlier in the paper (e.g., in the introduction) rather than thrown in at the very end.

Figure 1 is unreadable when printed out in B&W.

The stimuli generation in Experiment 1 seemed odd. Why not directly generate the functions via generating random coefficients rather than sample random points and fit a polynomial? If there was a good reason for this, explaining it in the paper would be helpful.

“Model-free results” -> “Human results”

At times, the framing gets into hairy theoretical territory as to the level of analysis of the paper, but at other times, it is fine. Examples of troubling language include “humans might be following a GP,” and “humans may use GP.” This ventures dangerously into the process level. The authors have not shown any evidence that the cognitive mechanism is “GP-like” whatever that might mean. An example of language at the appropriate level of analysis is “Results of optimization tasks suggest that human clicks during search for a maximum of a 1D function can be predicted by a Gaussian processes [sic] model.” (Perhaps “a Gaussian processes model” should be “a Gaussian process model” instead?).

I think the authors should report the rank-order correlation of the DVs for people and each optimization method (where the correlation is over the different function types).

I don’t understand why different tolerance levels were used for different optimization algorithms, although I do not see how it would prejudice their results in favor of their conclusion.

Figure 3 is extremely difficult to read when printed out in B&W.

Text is too small on Figure 4.

Results of Experiment 2 & 3: “Interestingly, on Deg5 functions, GP is closer to human clicks than the actual function (signed-rank test, p = 0.053) implying that GP captures clicks well in this case.” Would this hold up after correcting for multiple comparisons? (you tested Deg2 and Deg3 as well).

Experiments 2 & 4 are essentially function learning experiments.

Figure 5 is pretty difficult to read when printed out in B&W.

Experiment 5: Why were subjects always asked to guess the value of f(200) rather than a few different values? Also for the active extrapolation in this case, why not ask to find the value of the point right next to f(200) and guess basically that value?
Another control method for the interpolation and extrapolation methods that would have been nice to see how people and GPs relate to is estimating the parameters of a polynomial of degree up to m (and vary m).

Some recent work on optimal foraging presented at last year’s NIPS was Abbott, Austerweil, & Griffiths (2011) and a GP account might relate to the model used by Hills et al. (2011).
Summary: A strong paper, although I would have liked to see more integration with previous work and more ecologically valid stimuli/cover story.

Submitted by Assigned_Reviewer_9

This paper compared human behavior on several active search tasks to a set of established active search algorithms, and found that the Bayesian optimization algorithms combined with GP prior are better in capturing human data.

Quality:

In general, the paper is technically sound. Nevertheless, one major problem I have regarding their method is that the forms of the algorithms are not very proper (they are too simple) to solve the more complicated tasks as given to the humans. For example, the algorithms do not incorporate the step cost of querying a location whereas the humans do. The humans are essentially solving a composite sampling/stopping problem, whereas the algorithms separate sampling and stopping, and use seemingly arbitrary stopping rules that are not sensitive to the objective (i.e. higher hit with smaller function calls). This might not affect the general conclusion of this paper (that BO algorithms with GP can capture human search well), but it needs to be addressed to make the comparison between human and algorithms really fair. As I see it, Figure 4e actually implies that if the algorithms use a more sensible, joint policy for search and stopping, they could have achieved the same performance as people. I see that the authors partially addressed this issue in their discussion, but I think this issue needs more elaboration.

Another problem I have is that most of the BO algorithms use different methods than the non-Bayesian algorithms to decide where to query the next location. How do the authors separate the contribution of GP and the sampling policy? For example, GP-UCB looks pretty good. Is it because of GP learning or the UCB sampling policy?

Clarity:

Overall, the text is pretty well-written, but the figures lack clear descriptions (see list of specific questions).

Originality:

This paper is original.

Significance:

I have problems with the method as stated above, but I think the paper poses an important and interesting question that opens many future directions, along with unique, interesting experimental data. It also considered a pretty complete set of the most well-known search algorithms.

Minor comments:

- Page 3 1st paragraph: shouldn't it be 25 functions and 23 subjects?
- Figure 3: I don't see MPI intersect the box (as claimed in the text)..
- Page 4: Which second-order measure captures the order of the query sequence? I guess I'm not clear how "mean shortest distance" is calculated.
- What are the histograms in several plots in Figure 6?


Summary: A good paper. It is original and addresses important, interesting question. Some problems with the current form of their approach/method.
Author Feedback

Author rebuttal: Thanks for your constructive comments.

R#1: The amount of jitter depends on the max and min y values of each function (~50% of the range).
Technically, for each trial, we set: ylim([fmin-offset fmax+offset]) where fmin,
fmax are function extrema and offset is a random value in [0 (fmax-fmin)/2]. We admit that jittering does not completely remove
the spatial information and may give a small amount of information to humans. But note that many of the algorithms employed here do not
allow explicit incorporation of y bounds in their formulations, so even if we tried to give them the y bounds, they would just ignore them.
Hence, this may be a a promising research direction. We will discuss this.

Regarding giving K points and checking the K+1 point: For some algorithms (e.g., GA, DIRECT), it is not possible to do
this, while GPs naturally allow this (by replacing the selection function with human data; Exp. 7). However, some variations of
this idea may be possible. Agreement in search step sizes could indicate similarity in search strategies between humans
and algorithms. For example, random and fminBnd algorithms have idiosyncratic behaviors in step sizes, making them unlikely to be
employed by humans.


Exp 6: Subjects had to minimize the cost (ymax - their best y) so it was important for them to choose the maximum
location ASAP. In the absence of such costs or different ones, perhaps subjects would choose to see the max GP STD point first. The best
strategy thus is to choose max GP Mean in both selections. The idea behind letting subjects to choose a second point was to test whether
they follow a GP. In Exp. 6 (page 7), we aimed to contrast two hypotheses, the one based on the max STD (null hypothesis) and the
other based on max GP Mean. We will rephrase this section.

We will change 'gradient' to 'gradient direction'.

R#2. Thanks. Your points on abstract nature of our stimuli are well taken. The design was intentional to separate the features from employed strategies.
We will elaborate more on this.

The main reason behind not exactly following function learning tasks is because it is hard to teach subjects non-linear functions of higher
complexity. Some preliminary results in another version of our tasks where the subject was to estimate the length of a line from the length
of another line (y=f(x)) shown to them simultaneously (pairs over time), indicates that humans might be using GP there as well.

There is basically no big difference in both stimuli generation approaches. We will mention this.

Different xTol levels for some algorithms were used to control the stopping conditions. We swept the x tolerance of an algorithm and
measured the best performance in terms of function calls and hit rates for fair comparison of models vs. humans.

On multiple comparisons in Exp.2&3: Since we are comparing models over each function type separately, multiple comparisons should be done
over models (here GP vs. 4 models). Thus, dividing p-values by 4 works in favor of our hypothesis, making the difference between the real function and the
GP model statistically significant.

Exps. 2&4 are essentially function learning (FL) experiments but we wanted to show generalization of the theory over FL and active
learning. In extrapolation task, F(200) was used to align subjects at one point making the analysis easy and clean. Note that in Exp. 5,
subject had to choose one point first. We did not ask f(101)! because it is very close to f(100) making the task trivial.

R#3:Composite sampling/stopping: We will further clarify this point: both humans and model solve sampling+stopping. Algorithms were set
to terminate on their own, by setting parameters like xTol. We swept over these parameters and picked the values that best approached human performance
to make Fig. 2. So we did not stop the algorithms arbitrarily, they stopped on their own, like humans did. For step cost, we assume that
minimizing the number of function calls is a major objective of algorithm designers, otherwise nobody will use their algorithm. So it is implicitly
built into the algorithm design, though of course different algorithms weigh step cost vs accuracy and thoroughness differently, which makes for an
interesting variety of algorithms (e.g., GA makes lots of function calls but often avoids local minima that gradient descent can get stuck into).
Finding better joint policies for search and stopping could give rise to new algorithms; here we assumed that the applied maths community had
already explored this topic thoroughly, thus we used existing algorithms rather than trying to invent new ones.

Both GP and Selection criteria are important. We believe accuracy of BGO models is mainly due to the power of Gaussian processes which
progressively update the belief over the function shape. This is supported by the fact that GP parameters are constant across
selection criteria and further by our results in Exp. 7 (with no selection criteria).

We will fix your minor comments. There are 25 functions and 23 subjects. MPI intersects the box but the color coding is wrong.
Mean shortest distance: First for each model point, the closest human point is sought and then results are averaged over all model points.
The first 2nd-order measure captures sequence (point-wise distance i.e., 1st model-1st human, etc.). Histograms represent human clicked
locations (Fig. 6 panel 5).